## RESEARCH ARTICLE

# Risk factors for Buruli ulcer disease in Ghana: A matched case-control study in four selected endemic districts of Eastern and Oti Regions

**Mawuli Gohoho**[1,2,3☯*], **Samuel Adolf Bosoka**[1,2,4], **George Sarpong Agyemang**[1,2,5], **Sorengmen Amos Ziema**[1,2,6], **James Alorwu**[1,2,7], **Hudatu Ahmed**[1], **Christian Atsu Gohoho**[1,8], **Isaac Annobil**[3], **Nana Konama Kotey**[9], **John Owusu Gyapong**[10,11☯]

1 Fred N. Binka School of Public Health, University of Health and Allied Sciences, Hohoe, Volta Region, Ghana, 2 Ghana Field Epidemiology and Laboratory Training Programme, Accra, Ghana, 3 Jasikan Municipal Health Directorate, Ghana Health Service, Jasikan, Oti Region, Ghana, 4 Disease Surveillance Unit, Volta Regional Health Directorate, Ghana Health Service, Ho, Volta Region, Ghana, 5 Ahafo Regional Health Directorate, Ghana Health Service, Hwidiem, Ahafo Region, Ghana, 6 Department of Health Information and Records Management, Ho Teaching Hospital, Ministry of Health, Ho, Volta Region, Ghana, 7 Tumu Municipal Hospital, Ghana Health Service, Tumu, Upper West Region, Ghana, 8 Neglected Tropical Diseases Programme Unit, Volta Regional Health Directorate, Ghana Health Service, Ho, Volta Region, Ghana, 9 National Buruli Ulcer Control Programme, Ghana Health Service, Accra, Ghana, 10 Centre for Neglected Tropical Diseases Research, Institute of Health Research, University of Health and Allied Sciences, Ho, Volta Region, Ghana, 11 African Research Universities Alliance, Legon, Accra, Ghana

☯ These authors contributed equally to this work.

\* mawuli.gohoho@ghs.gov.gh

## Abstract

### Background

Buruli ulcer disease (BUD) remains a poorly understood neglected tropical disease (NTD). The 2021–2030 WHO NTD Roadmap prioritises addressing knowledge gaps in BUD transmission and calls for the need to better understand the factors contributing to disease occurrence. In Ghana, reported BUD cases declined from over 600 in 2018 to 81 in 2023. While previous case-control studies in Ghana have used a 1:1 matching ratio, this study examined potential risk factors for BUD in four endemic districts using an improved methodological approach.

### Methods

A community-based 1:2 matched case-control study was conducted in four BUD-endemic districts (Akwapim South, Akwapim North-Okere, Jasikan, and Biakoye) in the Eastern and Oti Regions of Ghana. Seventy (70) BUD cases and 140 community controls were recruited and matched by age (±5 years), sex, and place of residence. Data on socio-demographic, behavioural, environmental, water use, and injury management factors were collected by trained research assistants using semi-structured questionnaires designed with KoboCollect. Multivariable conditional

**Data availability statement:** All data analysed during this study are included in this article and its Supporting information files. The anonymised dataset is provided as Supporting information (S1 Dataset).

**Funding:** The author(s) received no specific funding for this work.

**Competing interests:** The authors have declared that no competing interests exist.

logistic regression produced adjusted odds ratios (aORs) with 95% confidence intervals at $p < 0.05$.

## Results

In the multivariable analysis, farming without adequate protective clothing (aOR = 3.02, 95% CI: 1.03–8.89) and living near waterbodies (aOR = 4.45, 95% CI: 1.46–13.55) were associated with increased odds of BUD. Being married (aOR = 0.32, 95% CI: 0.13–0.78) and applying alcohol to injury sustained (aOR = 0.17, 95% CI: 0.03–0.83) reduced the odds of BUD.

## Conclusion

Farming without adequate protective clothing and proximity to waterbodies were the main risk factors for BUD in endemic districts in Ghana. In contrast, being married and practising injury care using alcohol appeared protective. The Ghana Health Service should promote the consistent use of protective clothing during agricultural activities, raise awareness among communities living near waterbodies, and encourage proper injury care practices to reduce the risk of BUD.

### Author summary

Buruli ulcer disease is a bacterial infection that causes chronic skin ulcers and often leads to long-term disability if not treated early. Despite its public health impact, the exact mode of transmission remains unclear. This study was carried out in four districts in Ghana where Buruli ulcer is common, to better understand behaviours and environmental factors that may still be putting people at risk of infection. Using a 1:2 matched case-control approach, the study found that people who farmed without adequate protective clothing and those who lived close to waterbodies were more likely to develop BUD. On the other hand, individuals who were married and those who applied alcohol to injuries were less likely to get the disease. These findings support practical, community-level preventive strategies such as promoting the use of protective gears during farming and encouraging proper injury care, which could help reduce the burden of BUD in areas where the disease is common.

## Introduction

Buruli Ulcer Disease (BUD) is one of the 21 priority neglected tropical diseases (NTDs) recognised by the World Health Organization (WHO) [1]. It is caused by *Mycobacterium ulcerans*, a bacterium from the same family as the causative organisms of tuberculosis and leprosy [2]. This mycobacterium produces a unique toxin, mycolactone, which causes tissue damage and inhibits local immune responses,

thereby suppressing pain [3,4]. The disease typically manifests as ulcers that affect the skin and sometimes bone, and when treatment is delayed, it can result in permanent disfigurement and disability [2]. Lesions frequently occur on the limbs: 35% on the upper limbs, 55% on the lower limbs, and 10% on other parts of the body [5]. All age groups and sexes are affected, but about 50% of cases occur in children under 15 years [6,7].

Globally, over 30 countries with tropical, subtropical, and temperate climates have reported BUD, primarily in Africa, South America, and the Western Pacific [4,8]. From 2010 to 2017, over 23,000 cases were reported to the WHO from 16 countries, of which 14 were in the African Region [9]. In 2023, 1,952 suspected cases were reported from 12 countries, with 1,573 from Africa and 379 from the Western Pacific [8]. In Africa, estimated prevalence rates vary widely, from 30 cases per 100,000 in Ivory Coast to 250 per 100,000 in Cameroon [10–13].

In Ghana, the first possible BUD case was reported in 1971 [14]. By 1999, a nationwide assessment estimated a prevalence of 20.7 per 100,000 [3], with the Eastern and Volta-Oti regions reporting rates of 16.9 and 9.6 per 100,000, respectively [3]. Amansie West district recorded the highest prevalence at 150.8 per 100,000 [3]. From 2002 to 2023, Ghana reported 13,475 suspected cases to WHO [8].

Several epidemiological studies have identified socio-demographic characteristics, behavioural practices, environmental conditions, water use, and injury management as key determinants of BUD exposure and progression. These are shown in the conceptual framework (Fig 1), which highlights how multiple factors interact to influence BUD risk.

For example, residing near agricultural plantations, wooded areas, or water bodies increases BUD risk. Studies have emphasised the geographic clustering of cases [15–19]. Behavioural patterns such as farming in marshy areas [20,21],

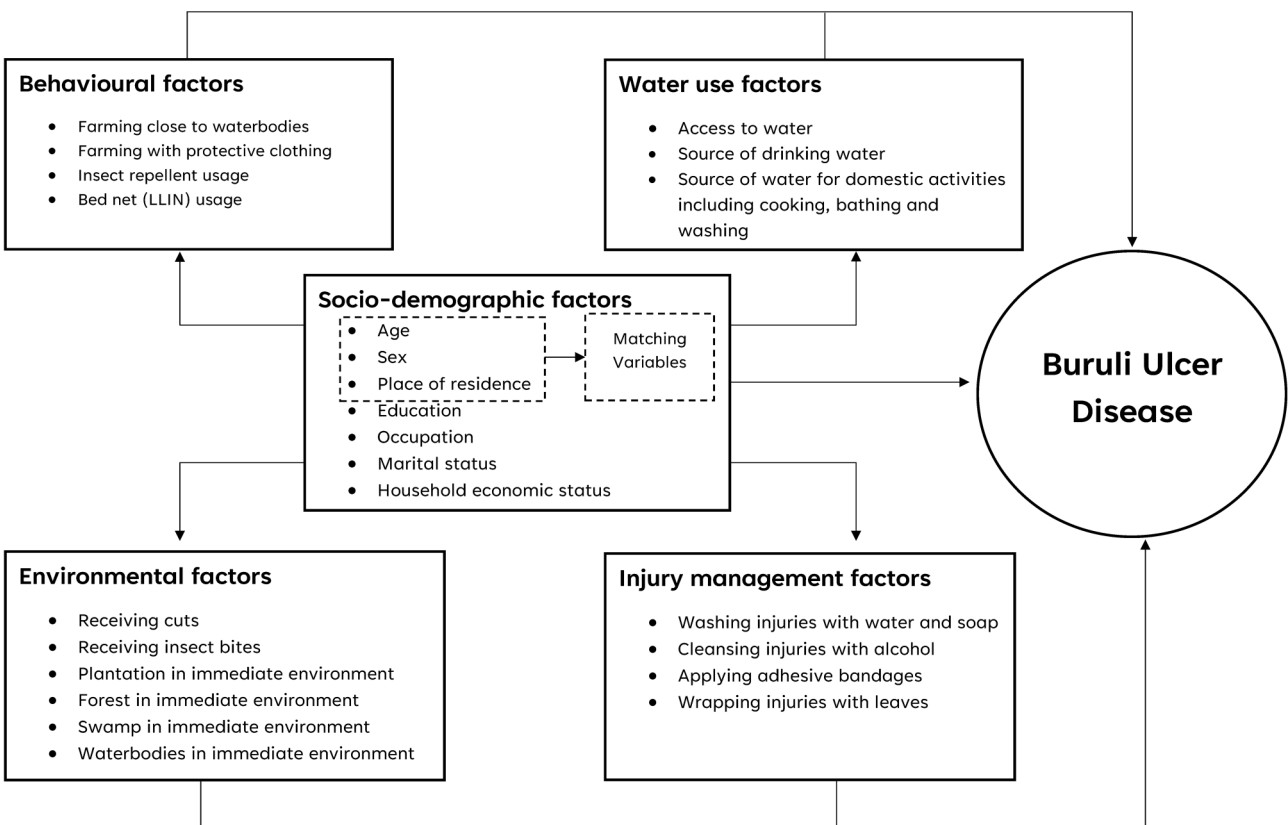

**Fig 1. Conceptual framework showing the interaction of multiple factors influencing BUD.**

exposure to insect bites [17,18,22,23], and using traditional remedies such as leaves for injury care increase the risk of contracting BUD [20]. Conversely, wearing protective clothing while farming reduces the risk [15,17,19,20,24], as do insect repellents and bed nets [16–18].

Environmental factors play an important role in the transmission of BUD. Living near wetlands increases the risk of contracting BUD [24]. Activities including wading in rivers, washing, or swimming in contaminated water bodies heighten exposure [19,24–26]. The bacterium is consistently detected in marshy ecosystems [20]. Use of unimproved water sources (such as ponds and swamps) for bathing, washing, or drinking have been strongly associated with the risk of BUD infection [6,16,17,21,23,26,27].

Injury management also influences BUD risk. Failure to clean or dress injuries sustained and using adhesive bandages increases susceptibility [16–18,24], whereas cleaning these injuries with soap and applying alcohol significantly reduce the risk [20,24]. Bathing with soap also lowers incidence [19] whereas proper injury care has been recommended as an essential intervention to minimizing the risk of BUD infection [28].

The exact mode of *Mycobacterium ulcerans* transmission remains unknown, although the disease clusters in communities near rivers, swamps, and wetlands [2,29,30]. Laboratory studies suggest aquatic insects, fish, plants, and terrestrial mammals could act as reservoirs. In Benin, *Mycobacterium ulcerans* was isolated from an aquatic water bug [31], supporting the theory of environmental reservoirs or vectors. These findings imply human infection may result from contact with contaminated water or materials [32]. Environmental disruptions such as dam building, deforestation, agriculture, and mining can disturb aquatic ecosystems, potentially facilitating the transmission of BUD to new areas [30,33].

The WHO NTD Roadmap 2021–2030 prioritises closing knowledge gaps in BUD transmission, highlighting the need to understand environmental and anthropogenic drivers [2]. Most Ghana-based studies have used a 1:1 matched case-control design to assess the risk factors associated with BUD [19,20,24,25]. However, increasing the control-to-case ratio can enhance statistical power, improve precision, and reduce standard error [34]. Reported BUD cases in Ghana have declined from over 600 in 2018 to 81 in 2023 [8]. Considering this decline, it is essential to re-assess risk factors in line with the WHO roadmap target to reduce case burdens through improved understanding of environmental dynamics. This study therefore examined the risk factors for BUD in four endemic districts in Ghana using a 1:2 matched case-control ratio.

## Methods

### Ethics statement

This study received ethical approval from the University of Health and Allied Sciences Research Ethics Committee (UHAS-REC A.8 [4] 20–21). Additional permission and administrative approval were obtained from the Health Directorates of the respective districts. Participation in the study was entirely voluntary, and written informed consent was obtained from all respondents. For child participants less than 18 years, written informed consent was obtained from a parent or guardian prior to inclusion. Participants were informed of their right to withdraw from the study at any time without any consequences. To protect privacy and ensure confidentiality, unique identification codes were assigned to each participant, and only these codes were used during data analysis. No personal identifiers were collected. Participants were not compensated for their involvement. Prior to data collection, the purpose, procedures, and duration of the study were clearly explained to all respondents. All information obtained was stored securely on a password-protected computer to maintain confidentiality.

### Study design

A community-based 1:2 matched case-control was designed in four BUD-endemic districts in the Eastern and Oti Regions of Ghana. Existing BUD cases reported through the BUD surveillance system were obtained from the Health Directorate of the respective districts and followed up to their respective communities. Controls were also selected from the same community as cases.

## Study area

The study was conducted in four BUD-endemic districts in Ghana: Akwapim South, Akwapim North-Okere, Jasikan, and Biakoye. Akwapim South and Akwapim North-Okere are located in the Eastern Region, while Jasikan and Biakoye are in the Oti Region (Fig 2). The Okere District was created and carved out of the Akwapim North District in 2018. However, for the purposes of this study, the "old Akwapim North" was maintained [35]. A common feature of the selected regions is Lake Volta, a reservoir formed by the Akosombo Dam on the lower Volta River in southern Ghana, which runs through both the Oti and Eastern Regions. The construction of the dam has however resulted in the displacement of many people and has had a significant implication on the local environment [36,37]. Report from the 2010 population and housing census indicate rivers and streams constitute the main sources of water for drinking 20.9% (range: 16.2% - 30.2%) and domestic activities 29.9% (range: 26.5% - 37.8%) respectively among the households in the four endemic districts [38].

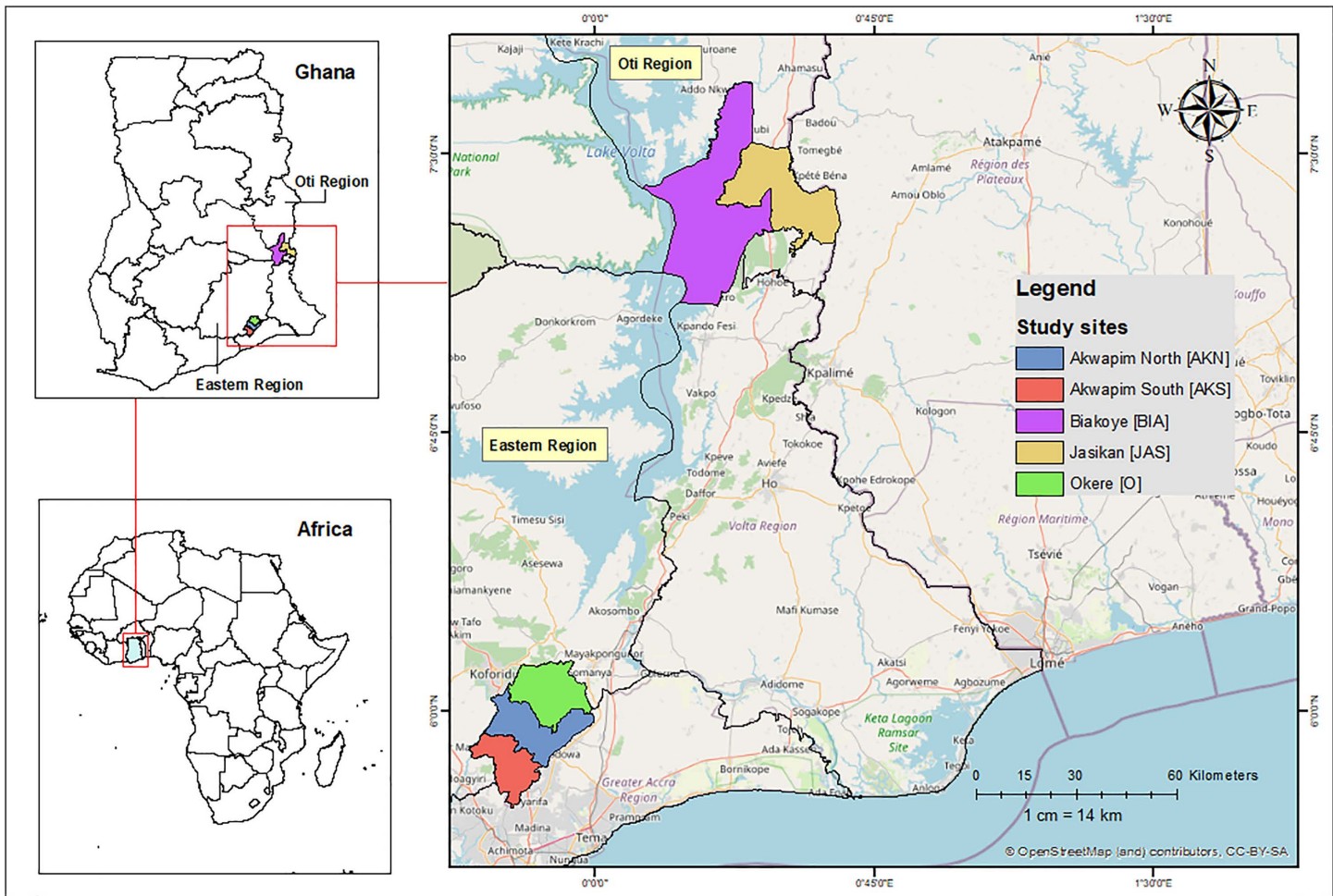

NB: The boundaries, names, and designations shown on this map do not imply the expression of any opinion regarding the legal status of any district, municipality, or region, nor do they indicate official endorsement or recognition of any administrative boundaries.

**Fig 2. Map of the study area showing the endemic districts in the Eastern and Oti Regions of Ghana.** (The base layer shapefile was obtained from GADM (https://gadm.org/download_country.html) and visualized using ArcMap 10.4. GADM permits academic use, including publication of maps in open-access journals under CC-BY licenses. License terms available at https://gadm.org/license.html.).

Crop farming 54.3% (range: 44.1% - 69.0%) is the predominant occupation of the people in these districts while others engage in the rearing of livestock 22.9% (range: 16.3% - 32.2%) [38]. Unlike other districts in Ghana which depend on passive reporting, these areas actively report on BUD cases to the National Buruli Ulcer Control Programme of the Ghana Health Service [39].

## Sample size

We calculated the sample size using the Fleiss method with the correction factor [40]. A z-score of 1.96 at a 95% confidence level and an 80% power were used. The districts reported a maximum of 40% usage of non-improved sources of water for domestic activities among households [38]. The minimum odds ratio for the association between cases and controls was set at 2.5 and a case-to-control ratio of 1:2 was used. Assuming a 10% non-response rate, we obtained a minimum sample size of 210 participants, made up of 70 cases and 140 controls.

## Case and control definition

**Case:** A probable case was defined as any person who had clinical signs of active BUD (including painless skin nodule, plaque or ulcer) or inactive BUD (healed lesions), resided in any of the BUD-endemic districts and was identified through the national Buruli ulcer surveillance system between 1 January 2018 and 31 March 2022. A confirmed case was any probable case with laboratory confirmation of *M.ulcerans* by IS2404-based polymerase chain reaction (PCR) analysis at Noguchi Memorial Institute for Medical Research, Ghana. Both probable and confirmed cases identified were included in this study.

 **Control:** An eligible control was defined as any person without any history or clinical symptoms of BUD who lived in any of the BUD-endemic districts between January 2018 to March 2022. Two controls were randomly selected and matched to each case by age (±5 years), sex, and place of residence (either the home of the case or a neighbouring home in the same community).

## Laboratory confirmation

All BUD cases recruited for this study were identified through the existing national Buruli Ulcer surveillance system. In Ghana, samples are collected by trained healthcare workers following standard operating procedures. Samples are placed in sterile transport media and stored at +4°C before being transported under cold chain conditions to the Noguchi Memorial Institute for Medical Research, Ghana. Laboratory confirmation was performed using the IS2404-based PCR assay, the standard for BUD diagnosis, following the protocol described in a previous Ghanaian study [41].

## Data collection

BUD cases for the study were recruited from the BUD line-list maintained by the Health Directorates in the four districts. These cases were systematically sampled from the sampling frame obtained from the Health Directorates. Because clinical and lesion characteristics were drawn from the surveillance line-list, they were based on routinely collected information documented by health workers at the time of diagnosis rather than direct assessment by the research assistants.

 Community health workers assisted in participant tracing to reduce selection bias. Trained research assistants visited the homes and communities of the cases to seek consent for their participation in the study. Amidst the COVID-19 pandemic, the research assistants were provided with face masks and hand sanitizers to minimize the risk of person-to-person transmission. After explaining the purpose of the study to the participants, two controls were selected from either the same household or a neighbouring household within the same community as the case.

 Upon obtaining informed consent from both cases and controls, a close-ended questionnaire designed using KoboCollect was administered to each participant in English or their local language to obtain information on socio-demographic,

behavioural, water use, environmental, and injury management factors. Cases were instructed to respond based on experiences in the year preceding the onset of BUD, while controls referred to the year preceding the interview. Research assistants received training to ensure consistent data collection. To minimise selection bias, community health workers and volunteers supported the tracing of cases and the identification of matched controls from the same or neighbouring households.

## Data management and analysis

The electronic responses were exported and analysed using STATA 16 (StataCorp LLC, College Station, Texas, USA). BUD status (case/control) was the outcome variable, while socio-demographic, behavioural, water use, environmental, and injury management factors served as the independent variables. Principal Component Analysis (PCA) was used to evaluate the household economic status of the respondents using 21 variables including household assets and sources of water supply. The Kaiser-Meyer-Olkin measure of sampling adequacy and Bartlett test of sphericity were first applied to determine whether the 21 variables contained sufficient collinearity to warrant use of PCA. The Kaiser-Meyer-Olkin measure was 0.624 and the Bartlett test was significant ($p < 0.001$) confirming the datasets amenability to PCA. Component scores from the PCA were combined to generate a composite wealth index, which was divided into quintiles and further categorised into three groups (low, average, and high household economic status) for subsequent analysis. Variable descriptions are provided in S1 File.

Cases and controls were matched during the design phase based on age, sex, and place of residence. To account for this matching during analysis, conditional logistic regression was used. Univariate analysis was conducted to calculate crude odds ratios (cOR) at 95% confidence intervals to assess the strength of association between the independent variables and BUD status. Variables with p-values <0.05 from the univariate analysis were included in the multivariable model. All variables that remained significant in the multivariable conditional logistic regression were considered predictors of BUD with a p-value <0.05. We assessed the model's goodness-of-fit using the likelihood ratio chi-square test.

## Results

### Clinical characteristics of BUD cases

Of the 70 cases, 46 (65.7%) were PCR-confirmed positive and the remaining cases were probable. Fifty-nine (84.3%) of the 70 cases had active lesions whereas 11 (15.7%) were in their inactive forms (healed lesions). Specifically, for the active cases, more than half [45 (64.3%)] of the cases were in their ulcerative forms. Of the 45 cases with ulcers, 32 (71.1%) were PCR-confirmed positive and the remaining 13 (28.9%) were probable cases (S1 Table). For cases with healed lesions, the median duration since healing was 9 months (IQR (interquartile range): 5–19). Most of the cases [39 (55.7%)] presented with category II lesions, followed by category III lesions [18 (25.7%)]. Almost all cases [67 (95.7%)] had lesions on their lower limbs, while 3 (4.3%) had lesions on their upper limbs. None of the cases had lesions on their head and neck, abdomen, fingers or other body parts. (Table 1).

### Socio-demographic characteristics

Overall, 70 (33.3%) cases and 140 (66.7%) controls were matched by sex, age, and community in the selected endemic districts in Ghana. Among the 70 case participants, 42 (60%) were males and 28 (40%) were females. The median age of the cases was 50 years (IQR: 3–91), with the majority [60 (85.7%)] aged 25 years and above. Thirty-three cases (47.2%) resided within the Jasikan Municipality. Among the cases, 59 (84.3%) were Christians, 36 (51.4%) had primary education, and 30 (42.9%) were farmers. Married persons accounted for 29 (41.4%) of the cases. Of the 29 married cases, 22 (75.9%) were PCR-confirmed positive and the remaining 7 (24.1%) were probable cases (S1 Table). In addition, 27 (38.6%) of the cases had low household economic status (Table 2).

**Table 1. Clinical and laboratory characteristics of BUD cases.**

| Variables | BUD cases n (%) | | Variables | BUD cases n (%) | |
|---|---|---|---|---|---|
| | 70 | (100.0%) | | 70 | (100.0%) |
| **Clinical forms** | | | **Location of lesion** | | |
| Nodules | 4 | (5.7) | Lower limb | 67 | (95.7) |
| Plaque | 1 | (1.4) | Upper limb | 3 | (4.3) |
| Oedema | 9 | (12.9) | Head and trunk | 0 | (0.0) |
| Ulcer | 45 | (64.3) | Abdomen | 0 | (0.0) |
| Healed | 11 | (15.7) | | | |
| **Duration of healed lesion (n = 11)** | | | **Case classification** | | |
| Median (IQR) | 9 (5-19) | | PCR-confirmed positive | 46 | (65.7) |
| | | | Probable | 24 | (34.3) |
| **Lesion category** | | | | | |
| Category I (< 5 cm) | 13 | (18.6) | | | |
| Category II (5 – 15 cm) | 39 | (55.7) | | | |
| Category III (>15 cm) | 18 | (25.7) | | | |

Among the 140 community controls, 84 (60%) were males and 56 (40%) were females. The median age of the controls was 50 years (IQR: 6–90), with most [120 (85.7%)] aged 25 years and above. Sixty-six controls (47.2%) resided within the Jasikan Municipality. One hundred and nineteen controls (85%) were Christians, 92 (65.7%) had primary education, and 64 (45.7%) were farmers. Married persons accounted for 86 (61.4%) of the controls. Assessment of household economic status showed that 59 (42.1%) of the controls had high household economic status (Table 2).

## Univariate analysis

**Socio-demographic factors:** Individuals with primary education [cOR = 0.24, 95% CI: 0.09–0.63] and those with secondary or higher education [cOR = 0.33, 95% CI: 0.12–0.96] had significantly lower odds of BUD compared to individuals without formal education. Married individuals also had significantly lower odds of BUD compared to those not married [cOR = 0.39, 95% CI: 0.20–0.74]. Although households with high economic status had reduced odds of BUD compared to those with low economic status [cOR = 0.87, 95% CI: 0.41–1.88], this association was not statistically significant (Table 2).

**Behavioural factors:** Individuals who farmed near waterbodies had higher odds of BUD compared to those who did not [cOR = 2.52, 95% CI: 1.11–5.70]. Similarly, those who farmed with inadequate protective clothing had nearly three times the odds of BUD compared to individuals not engaged in farming [cOR = 2.93, 95% CI: 1.27–6.81]. No significant association was observed between BUD and the use of insect repellents [cOR = 0.87, 95% CI: 0.47–1.63] or bed net use [cOR = 0.57, 95% CI: 0.28–1.15] (Table 3).

**Water use factors:** Both cases and controls accessed various water sources, including pipe-borne or protected wells, unprotected wells, rivers or streams, ponds, rainwater, and bottled water. Individuals who used unimproved water sources for drinking [cOR = 2.73, 95% CI: 0.65–11.51], cooking [cOR = 3.36, 95% CI: 0.83–13.55], or bathing [cOR = 1.93, 95% CI: 0.86–6.61] had higher odds of BUD compared to those who did not use such sources, though the associations were not statistically significant (Table 4).

**Environmental factors:** Individuals with waterbodies nearby had significantly higher odds of BUD, being three times more likely to contract the disease compared to those without such features in their immediate environment [cOR = 3.05, 95% CI: 1.41–6.60]. Similarly, those who reported insect bites during farming had higher odds of BUD compared to those who did not [cOR = 1.45, 95% CI: 0.68 – 3.07], though the association was not statistically significant (Table 5).

**Table 2. Univariate analysis of socio-demographic variables for BUD.**

| Variable | Cases n (%) | | Controls n (%) | | cOR (95% CI) | p-value |
|---|---|---|---|---|---|---|
| **Sex** | | | | | Matching variable | |
| Male | 42 | (60.0) | 84 | (60.0) | | |
| Female | 28 | (40.0) | 56 | (40.0) | | |
| **Age in years** | | | | | | |
| Median (IQR) | 50 (3-91) | | 50 (6-90) | | | |
| **Age group (years)** | | | | | Matching variable | |
| <15 | 7 | (10.0) | 11 | (7.9) | | |
| 15-24 | 3 | (4.3) | 9 | (6.4) | | |
| ≥25 | 60 | (85.7) | 120 | (85.7) | | |
| **District of residence (Geographical location)** | | | | | Matching variable | |
| Akwapim North-Okere | 11 | (15.7) | 22 | (15.7) | | |
| Akwapim South | 14 | (20.0) | 28 | (20.0) | | |
| Biakoye | 12 | (17.1) | 24 | (17.1) | | |
| Jasikan | 33 | (47.2) | 66 | (47.2) | | |
| **Religious affiliation** | | | | | | |
| Islam | 11 | (15.7) | 21 | (15.0) | Reference | |
| Christianity | 59 | (84.3) | 119 | (85.0) | 0.88 (0.27 – 2.89) | 0.838 |
| **Educational Level** | | | | | | |
| No formal education | 15 | (21.4) | 10 | (7.1) | Reference | |
| Primary | 36 | (51.4) | 92 | (65.7) | **0.24 (0.09-0.63)** | **0.004\*** |
| Secondary or higher | 19 | (27.1) | 38 | (27.1) | **0.33 (0.12 – 0.96)** | **0.041\*** |
| **Occupation** | | | | | | |
| Trader | 15 | (21.4) | 31 | (22.1) | Reference | |
| Farmer | 30 | (42.9) | 64 | (45.7) | 0.95 (0.38 – 2.34) | 0.910 |
| Student | 8 | (11.4) | 21 | (15.0) | 0.24 (0.02 – 2.49) | 0.235 |
| Artisan | 3 | (4.3) | 6 | (4.3) | 1.18 (0.26 – 5.30) | 0.832 |
| Fisher | 1 | (1.4) | 4 | (2.9) | 0.47 (0.04 – 6.13) | 0.564 |
| others[α] | 13 | (18.6) | 14 | (10.0) | 2.06 (0.71 – 5.96) | 0.181 |
| **Marital Status** | | | | | | |
| Non-married[β] | 41 | (58.6) | 54 | (38.6) | Reference | |
| Married | 29 | (41.4) | 86 | (61.4) | **0.39 (0.20 - 0.74)** | **0.004\*** |
| **Household Economic Status** | | | | | | |
| Low | 27 | (38.6) | 57 | (40.7) | Reference | |
| Average | 18 | (25.7) | 24 | (17.1) | 1.66 (0.72 – 3.84) | 0.235 |
| High | 25 | (35.7) | 59 | (42.1) | 0.87 (0.41 – 1.88) | 0.729 |

[α]other occupations include; not engaged in work (9), driver (3), civil servant (6), gardener (2), painter/hairdresser/others (7).

[β]Non-married include single, divorced and widowed.

cOR – Crude Odds Ratio, CI – Confidence Interval, IQR – Interquartile range.

\* Statistically significant at p<0.05.

**Injury management factors:** Individuals with a visible Bacillus Calmette-Guérin (BCG) scar had higher odds of BUD [cOR = 1.41, 95% CI: 0.63–3.18], though the association was not statistically significant. Individuals with a family history of BUD had significantly lower odds of contracting BUD [cOR = 0.33, 95% CI: 0.16–0.71]. Regarding injury management, individuals who managed injuries sustained with soap and water [cOR = 0.25, 95% CI: 0.09–0.71] or cleaned them with

**Table 3. Univariate analysis of behavioural variables for BUD.**

| Variable | Cases n (%) | | Controls n (%) | | cOR (95% CI) | p-value |
|---|---|---|---|---|---|---|
| **Participate in Farming** | | | | | | |
| Do not farm | 20 | (28.6) | 49 | (35.0) | Reference | |
| Farm | 50 | (71.4) | 91 | (65.0) | 1.47 (0.72 – 3.01) | 0.291 |
| **Farming close to waterbodies** | | | | | | |
| Do not farm | 20 | (28.6) | 49 | (35.0) | Reference | |
| Farm near waterbodies | 37 | (52.9) | 45 | (32.1) | **2.52 (1.11 – 5.70)** | **0.027*** |
| Farm away from waterbodies | 13 | (18.6) | 46 | (32.9) | 0.76 (0.32 – 1.80) | 0.527 |
| **Farming with protective clothing** | | | | | | |
| Do not farm | 20 | (28.6) | 49 | (35.0) | Reference | |
| Farm with inadequate protective clothing[α] | 30 | (42.9) | 31 | (22.1) | **2.93 (1.27 – 6.81)** | **0.012*** |
| Farm with adequate protective clothing[β] | 20 | (28.6) | 60 | (42.9) | 0.75 (0.32 – 1.72) | 0.494 |
| **Insect repellent usage** | | | | | | |
| No | 31 | (44.3) | 58 | (41.4) | Reference | |
| Yes | 39 | (55.7) | 82 | (58.6) | 0.87 (0.47 – 1.63) | 0.670 |
| **LLIN[δ] usage** | | | | | | |
| No | 25 | (35.7) | 37 | (26.4) | Reference | |
| Yes | 45 | (64.3) | 103 | (73.6) | 0.57 (0.28 – 1.15) | 0.119 |

[α]Farming with less than 3 of these clothing; long sleeves, trousers and closed shoes.

[β]Farming with all 3 or more of these clothing; long sleeves, trousers and closed shoes.

[δ]Long-lasting insecticidal net.

* Statistically significant at p < 0.05.

cOR – Crude Odds Ratio, CI – Confidence Interval.

alcohol [cOR = 0.24, 95% CI: 0.08–0.76] had significantly reduced odds of BUD. In contrast, those who wrapped their injuries with leaves had over three times higher odds of BUD compared to those who did not [cOR = 3.35, 95% CI: 1.54–7.26] (Table 6).

## Multivariable analysis

Farming with inadequate protective clothing [aOR =3.02, 95% CI: 1.03-8.89] and the presence of waterbodies in an individual's immediate environment [aOR =4.45, 95% CI: 1.46-13.55] were risk factors associated with BUD. However, being married [aOR =0.32, 95% CI: 0.13-0.78], managing injuries sustained with alcohol [aOR =0.17, 95% CI: 0.03-0.83] were found to be protective factors (Table 7).

## Discussion

In this community-based matched case-control study, farming without adequate protective clothing and proximity to waterbodies were associated with increased odds of BUD in four endemic districts of Ghana, while being married and alcohol application on injuries appeared protective. These findings are discussed in relation to previous studies from West Africa and elsewhere, with attention to their implications for BUD surveillance and control strategies in Ghana, particularly within the framework of the WHO 2021–2030 NTD Roadmap [2].

In this study, most of cases presented at the ulcerative stage, with majority of lesions occurring on the lower limbs and nearly three-quarters of the lesions were identified when small in size and in their early stages (Category I and II). Similar findings were reported in Benin [42] but contrary to another study in Nigeria [43]. This may reflect

**Table 4. Univariate analysis of water use variables for BUD.**

| Variable | Cases n (%) | | Controls n (%) | | cOR (95% CI) | p-value |
|---|---|---|---|---|---|---|
| **Water access** | | | | | | |
| **Pipe borne/protected well** | | | | | | |
| No | 12 | (17.1) | 19 | (13.6) | Reference | |
| Yes | 58 | (82.9) | 121 | (86.4) | 0.45 (0.12 - 2.74) | 0.249 |
| **Unprotected well** | | | | | | |
| No | 68 | (97.1) | 131 | (93.6) | Reference | |
| Yes | 2 | (2.9) | 9 | (6.4) | 0.38 (0.08 - 1.97) | 0.252 |
| **River/stream** | | | | | | |
| No | 42 | (60.0) | 77 | (55.0) | Reference | |
| Yes | 28 | (40.0) | 63 | (45.0) | 0.67 (0.29 - 1.53) | 0.343 |
| **Pond/dug/out/lake/dam** | | | | | | |
| No | 67 | (95.7) | 132 | (94.3) | Reference | |
| Yes | 3 | (4.3) | 8 | (5.7) | 0.71 (0.17 – 3.0) | 0.639 |
| **Rain water** | | | | | | |
| No | 26 | (37.1) | 58 | (41.4) | Reference | |
| Yes | 44 | (62.9) | 82 | (58.6) | 1.3 (0.64 - 2.63) | 0.474 |
| **Sachet/bottled water** | | | | | | |
| No | 48 | (68.6) | 89 | (63.6) | Reference | |
| Yes | 22 | (31.4) | 51 | (36.4) | 0.69 (0.32 - 1.52) | 0.360 |
| **Primary water source for drinking** | | | | | | |
| Improved source[a]* | 57 | (81.4) | 120 | (85.7) | Reference | |
| Unimproved source[β] | 13 | (18.6) | 20 | (14.3) | 2.73 (0.65 – 11.51) | 0.171 |
| **Primary water source for cooking** | | | | | | |
| Improved source[a] | 53 | (75.7) | 114 | (81.4) | Reference | |
| Unimproved source[β] | 17 | (24.3) | 26 | (18.6) | 3.36 (0.83 – 13.55) | 0.088 |
| **Primary water source for bathing** | | | | | | |
| Improved source[a] | 49 | (70.0) | 107 | (76.4) | Reference | |
| Unimproved source[β] | 21 | (30.0) | 33 | (23.6) | 2.31 (0.81 – 6.61) | 0.120 |
| **Local soap for bathing** | | | | | | |
| No | 17 | (24.3) | 46 | (32.9) | Reference | |
| Yes | 53 | (75.7) | 94 | (67.1) | 1.93 (0.86 – 4.36) | 0.113 |
| **Primary water source for washing** | | | | | | |
| Improved source[a] | 49 | (70.0) | 106 | (75.7) | Reference | |
| Unimproved source[β] | 21 | (30.0) | 34 | (24.3) | 2.00 (0.73 – 5.47) | 0.177 |
| **Local soap for washing** | | | | | | |
| No | 19 | (27.1) | 38 | (27.1) | Reference | |
| Yes | 51 | (72.9) | 102 | (72.9) | 1.00 (0.44 – 2.30) | 1.00 |

[a]Improved sources include piped borne water, rainwater, borehole/tube well.

+ Includes sachet/bottled water.

[β]Unimproved sources include unprotected well, river/stream and pond/dugout/lake/dam.

cOR – Crude Odds Ratio, CI – Confidence Interval.

**Table 5. Univariate analysis of environmental variables for BUD.**

| Variable | Cases n (%) | | Controls n (%) | | cOR (95% CI) | p-value |
|---|---|---|---|---|---|---|
| **Cuts during farming activities** | | | | | | |
| Do not farm | 20 | (28.6) | 49 | (35.0) | Reference | |
| Farm, never had cuts | 12 | (17.1) | 19 | (13.6) | 1.68 (0.63 – 4.44) | 0.298 |
| Farm, had cuts | 38 | (54.3) | 72 | (51.4) | 1.40 (0.65 – 2.99) | 0.389 |
| **Insect bites during farming activities** | | | | | | |
| Do not farm | 20 | (28.6) | 49 | (35.0) | Reference | |
| Farm, never bitten by insects | 10 | (14.3) | 17 | (12.1) | 1.55 (0.55 – 4.36) | 0.402 |
| Farm, bitten by insects | 40 | (57.1) | 74 | (52.9) | 1.45 (0.68 – 3.07) | 0.337 |
| **Plantation in immediate environment** | | | | | | |
| No | 32 | (45.7) | 71 | (50.7) | Reference | |
| Yes | 38 | (54.3) | 69 | (49.3) | 1.53 (0.65 – 3.57) | 0.325 |
| **Forest in immediate environment** | | | | | | |
| No | 42 | (60.0) | 81 | (57.9) | Reference | |
| Yes | 28 | (40.0) | 59 | (42.1) | 0.81 (0.33 – 1.98) | 0.651 |
| **Swamp in immediate environment** | | | | | | |
| No | 48 | (68.6) | 110 | (78.6) | Reference | |
| Yes | 22 | (31.4) | 30 | (21.4) | 2.39 (0.99 – 5.77) | 0.053 |
| **Waterbodies in immediate environment** | | | | | | |
| No | 31 | (44.3) | 86 | (61.4) | Reference | |
| Yes | 39 | (55.7) | 54 | (38.6) | **3.05 (1.41 – 6.60)** | **0.005*** |

* Statistically significant at p < 0.05.

cOR – Crude Odds Ratio, CI – Confidence Interval.

robust surveillance activities, with most cases detected before extensive tissue destruction, reducing complications and improving healing. Nonetheless, approximately one-quarter were diagnosed late, possibly due to transportation costs, loss of income, and other barriers [44–46]. To meet the WHO 2030 Roadmap milestones of reducing Category III lesions to below 10% and achieving more than 95% laboratory confirmation [2], Ghana must sustain active surveillance, ensure universal PCR testing, and expand access to rapid diagnostics at peripheral levels. For the fact that lesions remain heavily concentrated in the ulcerative form on the lower limbs, control strategies must integrate behavioural interventions such as the use of protective clothing during farming and improved wound hygiene practices. There is also the need for the National Buruli Ulcer Control Programme to periodically build capacity of community health workers to improve early case detection and referral, while improving decentralized treatment centres to remove barriers linked to transport and income loss.

Married individuals had lower odds of BUD than non-married. It is essential to note that marital status was not a matching variable, so the observed association may partly reflect residual confounding with other factors. Nonetheless, literature acknowledges health benefits of marriage, including economic, behavioural, and health advantages [47–49]. Spouses may monitor each other's health behaviours and discourage risky actions [50], which could possibly reduce injury and exposure to *Mycobacterium ulcerans*. Moreover, with better education and resources, partners may ensure timely healthcare access. However, a recent study in Ghana found no such association [20].

Farming with inadequate protective clothing was associated with increased odds of BUD. In this study, farmers who did not wear both long-sleeved upper clothing and long trousers, along with closed shoes, were classified as having inadequate protective clothing. These findings are consistent with previous studies in Ghana [19,20,24]. Due to hot weather, farmers may prefer lighter clothing during farming activities to maximize productivity [20]. This behaviour however

**Table 6. Univariate analysis of Injury management variables for BUD.**

| Variable | Cases n (%) | | Controls n (%) | | cOR (95% CI) | p-value |
|---|---|---|---|---|---|---|
| **BCG vaccination status[a]** | | | | | | |
| Not Vaccinated | 22 | (31.4) | 50 | (35.7) | Reference | |
| Vaccinated | 48 | (68.6) | 90 | (64.3) | 1.41 (0.63 – 3.18) | 0.407 |
| **Family history of BUD** | | | | | | |
| No | 59 | (84.3) | 93 | (66.4) | Reference | |
| Yes | 11 | (15.7) | 47 | (33.6) | **0.33 (0.16 – 0.71)** | **0.005*** |
| **Injury management** | | | | | | |
| **Managing injuries with water** | | | | | | |
| No | 31 | (44.3) | 72 | (51.4) | Reference | |
| Yes | 39 | (55.7) | 68 | (48.6) | 1.60 (0.76 – 3.38) | 0.214 |
| **Managing injuries with water and soap** | | | | | | |
| No | 64 | (91.4) | 108 | (77.1) | Reference | |
| Yes | 6 | (8.6) | 32 | (22.9) | **0.25 (0.09 – 0.71)** | **0.009*** |
| **Managing injuries with water and salt** | | | | | | |
| No | 44 | (62.9) | 88 | (62.9) | Reference | |
| Yes | 26 | (37.1) | 52 | (37.1) | 1.00 (0.45 – 2.30) | 1.00 |
| **Cleansing injuries with alcohol** | | | | | | |
| No | 63 | (90.0) | 110 | (78.6) | Reference | |
| Yes | 7 | (10.0) | 30 | (21.4) | **0.24 (0.08 – 0.76)** | **0.015*** |
| **Applying adhesive bandages** | | | | | | |
| No | 54 | (77.1) | 113 | (80.7) | Reference | |
| Yes | 16 | (22.9) | 27 | (19.3) | 1.29 (0.60 – 2.74) | 0.512 |
| **Wrapping injuries with leaves** | | | | | | |
| No | 44 | (62.9) | 113 | (80.7) | Reference | |
| Yes | 26 | (37.1) | 27 | (19.3) | **3.35 (1.54 – 7.26)** | **0.002*** |

[a] Vaccination status assessed based on the presence of BCG scar.

cOR – Crude Odds Ratio, CI – Confidence Interval.

* Statistically significant at p < 0.05.

**Table 7. Multivariable model of conditional logistic regression for risk factors associated with BUD.**

| Variable | aOR (95% CI) | p-value |
|---|---|---|
| Farming with inadequate protective clothing | 3.02 (1.03 - 8.89) | 0.045* |
| Waterbodies present in immediate environment | 4.45 (1.46 - 13.55) | 0.009* |
| Cleansing injuries with alcohol | 0.17 (0.03 - 0.83) | 0.029* |
| Being married | 0.32 (0.13 - 0.78) | 0.013* |

Model fit: McFadden's $R^2$ = 0.3775, LR $\chi^2$ = 58.05, p < 0.001.

aOR – Adjusted Odds Ratio, CI – Confidence Interval.

*Statistically significant at p < 0.05.

NB: After multivariable analysis, variables not statistically significant at p < 0.05 included educational level, family history of BUD, managing injuries with water and soap, and wrapping injuries with leaves. These variables were included in the analysis for completeness but did not show any significant associations with the outcome. Additionally, the variable 'farming close to waterbodies' was dropped from the final model due to collinearity with other farming-related variables.

increases skin exposure and disease risk. This is evident in lesion localisation, with over 95% on limbs. However, other studies found no significant protective effect from wearing protective clothing [16,17,23].

Though excluded from multivariable analysis due to collinearity, univariate analysis showed that farming away from rivers reduced BUD risk. Agricultural activity around waterbodies increases contact with moist environments. *Mycobacterium ulcerans* has been isolated from soil in BUD-endemic areas, particularly along the Densu river basin [51]. These pathogens may thrive in submerged decaying organic matter [1], facilitating transmission during water contact. Moreover, individuals living near waterbodies had increased odds of BUD. Studies have linked proximity to rivers, swamps, or wetlands with elevated BUD risk [17,19,21]. Aquatic environments may serve as reservoirs or breeding grounds for vectors [51–54]. Though findings are consistent with other studies, proximity alone is believed to be insufficient for transmission. The nature of interaction such as bathing, wading, farming or performing other activities that result in skin exposure or trauma may be important [25,55]. Minor skin trauma during such activities may provide portals of entry, supporting infection. From a One Health perspective, this demonstrates the interplay of human behaviour, environment, and pathogen ecology [1,56,57].

Despite no association found in this study between insect bites and BUD, other studies in Australia, Cameroon, Togo, and Ghana have shown otherwise [17,18,22–24]. Insects may act as vectors or reservoirs [32]. However, protective measures including bed nets and clothing may interrupt the transmission cycle. In addition, water access and primary sources for domestic use showed no significant association in both univariate and multivariable analyses. Similar findings were reported elsewhere [58,17,19,20,23]. Yet, Aiga and colleagues found a significant association with unimproved water sources. In 2010, two-fifth of the population drank from unimproved sources, but in this study, less than 15% of controls reported such use, suggesting improvement in improved water access [38]. No significant association was found between BCG vaccination and odds of contracting BUD. This is consistent with previous studies [19,23,24,59]. Most studies used presence of scar as proxy for vaccination, but scar-based assessment has a sensitivity range of 55–97% [60–62] and its does not equate to immunity, as absence may result from poor technique [63–65].

Again, use of leaves for injury management was associated with increased BUD risk. Similar findings were reported in Ghana and Cameroon [17,20,24]. Aquatic plants may harbour *Mycobacterium ulcerans* [66], and open wounds facilitate direct entry of pathogens [28]. However, application of alcohol on wounds appeared protective. Alcohol-based disinfectant is a broad-spectrum agent that can neutralise pathogens at points of skin breach [67]. In Ghana, methylated spirit is a common household alcohol-based disinfectant used as a first-aid treatment for open wounds. Since *Mycobacterium ulcerans* likely enters through minor injuries, alcohol application may help prevent colonisation and inhibit growth, thus reducing infection risk. However, their use is typically limited to initial cleansing, as they lack residual activity and may impair tissue healing if used excessively [68]. Though few direct studies link alcohol use to BUD prevention, general wound hygiene is recommended in endemic settings [67,69].

Overall, our findings corroborate the importance of protective measures during farming, avoiding contact with waterbodies, and promoting hygienic injury care practices such as alcohol application. The association of marital status with reduced BUD risk suggests potential pathways through improved health-seeking behaviours and social support. Until the exact mode of transmission is fully clarified, delivering public health information and continuously increasing awareness of these risk factors remain essential strategies for both the prevention and control of BUD. Embracing a One Health approach that integrates human, animal, and environmental health perspectives will be vital in identifying transmission pathways and developing comprehensive control strategies [1]. The generalizability of these findings however is likely limited to BUD-endemic districts in Ghana with similar environmental and agricultural profiles, such as those near rivers, swamps, or wetlands like Lake Volta, where about one-fifth of households rely on rivers for drinking water [38]. The associations with proximity to waterbodies and as well as farming practices may not apply to non-endemic regions or urban areas with greater access to improved water sources. Nonetheless, the protective effects of injury management practices, such as alcohol application, are likely broadly applicable to other settings where minor skin trauma is common.

It is worth noting that the typical limitations of case-control studies are relevant to this study and caution should be taken when interpreting these findings. One of such limitation is recall bias, which may have affected both cases and controls, particularly in their reporting of farming practices, use of protective clothing, and injury management. This may have led to under- or overestimation of associations with the risk of BUD. Blinding of research assistants to participant outcomes was also not feasible, though research assistants were trained to ensure accurate data collection and minimize bias. Although most cases were PCR-confirmed positive, less than one-third were probable cases obtained through the national Buruli ulcer surveillance system. This inclusion may introduce uncertainty, as many of these probable cases had lesions on the lower limbs, which could also arise from other causes. Nonetheless, health workers in the study districts had received training on case detection, diagnosis, and management through the National Buruli Ulcer Control Programme, as documented in the 2019 Annual Report of the Buruli Ulcer Control and Yaws Eradication Programme [1]. Nevertheless, we recommend that future studies in Ghana build on the design explored in this study but rely solely on PCR-confirmed cases to improve validity. Furthermore, there were challenges in reaching cases and identifying suitable controls. Due to logistical constraints and limited resources, accessing the target population proved difficult. BUD cases were dispersed across communities, making it challenging to locate and recruit matched controls. The difficulty in matching by age, sex, and residence further complicated control selection. These challenges were mitigated by relying on community health workers and volunteers to trace participants and using flexible scheduling. Research assistants also visited the homes and communities of cases directly, with controls selected from the same household or neighbourhood to facilitate participation.

## Conclusion

Our study identified farming without adequate protective clothing and living near waterbodies as main risk factors for BUD in endemic districts in Ghana. On the other hand, being married and applying alcohol over injuries appeared protective of BUD. The Ghana Health Service should encourage the consistent use of protective clothing during agricultural activities, educate communities living near waterbodies about their higher risk of BUD, and promote proper injury management practices as a simple and effective strategy to lower the risk of BUD. Spatial analysis research is needed to map BUD cases and determine whether their proximity to waterbodies or other environmental features increases risk, thereby identifying high-risk areas for targeted interventions.

## Supporting information

**S1 Table.** **Distribution of demographic and clinical characteristics by case classification.**
(DOCX)

**S1 Dataset.** **Anonymised dataset used for analysis.**
(XLS)

**S1 File.** **Variable dictionary describing variable names, definitions, and value labels.**
(PDF)

**S2 File.** **Translation of the abstract to French.**
(DOCX)

## Acknowledgments

This research was conducted as part of Mawuli Gohoho's MPhil in Applied Epidemiology programme at the Fred N. Binka School of Public Health, University of Health and Allied Sciences, Ghana. The authors acknowledge the Directors of Health Services in Akwapim North, Akwapim South, Okere, Jasikan, and Biakoye Districts for their support in facilitating

data collection. Appreciation is also extended to the trained research assistants, health workers, and Community-Based Surveillance Volunteers for their role in data collection, and to all study participants for their invaluable participation.

## Author contributions

**Conceptualization:** Mawuli Gohoho, John Owusu Gyapong.

**Data curation:** Mawuli Gohoho, Samuel Adolf Bosoka, John Owusu Gyapong.

**Formal analysis:** Mawuli Gohoho, Samuel Adolf Bosoka, George Sarpong Agyemang, Sorengmen Amos Ziema, John Owusu Gyapong.

**Investigation:** Mawuli Gohoho, John Owusu Gyapong.

**Methodology:** Mawuli Gohoho, John Owusu Gyapong.

**Visualization:** Mawuli Gohoho, James Alorwu, Hudatu Ahmed, Christian Atsu Gohoho, John Owusu Gyapong.

**Writing – original draft:** Mawuli Gohoho, John Owusu Gyapong.

**Writing – review & editing:** Mawuli Gohoho, Samuel Adolf Bosoka, George Sarpong Agyemang, Sorengmen Amos Ziema, James Alorwu, Hudatu Ahmed, Christian Atsu Gohoho, Isaac Annobil, Nana Konama Kotey, John Owusu Gyapong.

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
