## [Decision Letter · Decision Letter 0]

13 Aug 2025

Risk Factors for Buruli Ulcer Disease in Ghana: A Matched Case-Control Study in Four Selected Endemic Districts of Eastern and Oti Regions

Dear Dr. Gohoho,

Thank you for submitting your manuscript to PLOS Neglected Tropical Diseases. After careful consideration, we feel that it has merit but does not fully meet PLOS Neglected Tropical Diseases's publication criteria as it currently stands. Therefore, we invite you to submit a revised version of the manuscript that addresses the points raised during the review process.

Please submit your revised manuscript within 60 days Oct 12 2025 11:59PM. If you will need more time than this to complete your revisions, please reply to this message or contact the journal office at plosntds@plos.org. Please include the following items when submitting your revised manuscript:

We look forward to receiving your revised manuscript.

Kind regards,

Michael Marks

Academic Editor

Mathieu Picardeau

Section Editor

Shaden Kamhawi

co-Editor-in-Chief

Paul Brindley

co-Editor-in-Chief

**Journal Requirements:**

3) Some material included in your submission may be copyrighted. According to PLOSu2019s copyright policy, authors who use figures or other material (e.g., graphics, clipart, maps) from another author or copyright holder must demonstrate or obtain permission to publish this material under the Creative Commons Attribution 4.0 International (CC BY 4.0) License used by PLOS journals. Please closely review the details of PLOSu2019s copyright requirements here: PLOS Licenses and Copyright. If you need to request permissions from a copyright holder, you may use PLOS's Copyright Content Permission form.

Potential Copyright Issues:

i) Figure 2. Please (a) provide a direct link to the base layer of the map (i.e., the country or region border shape) and ensure this is also included in the figure legend; and (b) provide a link to the terms of use / license information for the base layer image or shapefile. We cannot publish proprietary or copyrighted maps (e.g. Google Maps, Mapquest) and the terms of use for your map base layer must be compatible with our CC BY 4.0 license.

4) Kindly revise your competing statement to align with the journal's style guidelines: 'The authors declare that there are no competing interests.'

**Reviewers' Comments:**

Reviewer's Responses to Questions

**Key Review Criteria Required for Acceptance?**

**Methods**

-Are the objectives of the study clearly articulated with a clear testable hypothesis stated?

-Is the study design appropriate to address the stated objectives?

-Is the population clearly described and appropriate for the hypothesis being tested?

-Is the sample size sufficient to ensure adequate power to address the hypothesis being tested?

-Were correct statistical analysis used to support conclusions?

-Are there concerns about ethical or regulatory requirements being met?

Reviewer #1: The objectives are clearly articulated with a population study identified based on WHO criteria

The sample was suffiscient in the number based on the method of sample estimation

Statistical analysis well done

The quality of patients as BU positive was not accurate since the majority was missclassified as BU patient based only on clinical aspects

Reviewer #2: (No Response)

**Results**

-Does the analysis presented match the analysis plan?

-Are the results clearly and completely presented?

-Are the figures (Tables, Images) of sufficient quality for clarity?

Reviewer #1: All results are clearly presented as well as the figure and table

However, the number of true BU patients is very low 27 of 70 as reported in the paper. This limit our ability to provide a good analysis of the results

Reviewer #2: (No Response)

**Conclusions**

-Are the conclusions supported by the data presented?

-Are the limitations of analysis clearly described?

-Do the authors discuss how these data can be helpful to advance our understanding of the topic under study?

-Is public health relevance addressed?

Reviewer #1: The study has stated clearly the limitation which are very critical and limit the usefull of the data provided.

This conclusion need to be taken with some restriction

Reviewer #2: (No Response)

**Editorial and Data Presentation Modifications?**

Reviewer #1: This study need a major revision

Reviewer #2: (No Response)

**Summary and General Comments**

Reviewer #1: In general, this is a good and relevance study since the author wouls address the understanding of the decline in the number of the BU cases with the associated risk factors in 4 endemics area of BU in Ghana. All sections of the paper are well designed and written. The methodology was clear and the approach of the field investigation was good. Unfortunately, the study introduce itself a criticial biais on the classification of the BU patients on whom the study is supporting to select the control matched by age and area of residence. This limitation could not help us to consider the results obtain and the conclusion of the study.

Reviewer #2: Buruli ulcer, a Neglected Tropical disease is known to be highly endemic in most West African countries including Ghana. Although, the mode of transmission is still unknown, the changing epidemiology of the disease necessitates the need to re-assess risk factors in line with the WHO roadmap target to reduce case burdens through improved understanding of environmental dynamics. The manuscript requires significant improvements in several areas. There are some recommendations to improve the quality of the manuscript.

Introduction

Line 78-79: modify to reflect the fact that permanent disfigurement and disability are likely the consequence of delayed treatment.

Line 133-134: only 81 cases reported in Ghana in 2023 yet authors had 70 cases in <1year period from 4 districts. Can the authors explain the discrepancy?

1. Authors maintained Buruli Ulcer disease as BUD. However, the term “BU” has been used throughout portions of the manuscript but not defined. Is BU used in the same context as BUD by the authors?

2.The use of the phrase “since then” to precede lines 90-91 is misleading. This portrays the meaning Buruli ulcer cases are still on the rise in Ghana, even currently. Meanwhile the number of reported cases from Ghana have steadily been on the decline since 2019. In addition, the reference [15] used is outdated.

It seems to suggest some sentences in the introduction are not complete, thus making it difficult to get the clear import of the message the authors are putting across. The first sentence on line 110 is incomplete. “Environmental factors are central”. It is not clear what the authors are saying by this sentence. Similarly, “Use of unimproved water sources (such as ponds and swamps) for bathing, washing, or drinking is strongly associated with BUD”. Do the authors by this sentence mean the risk of BUD infection or what?

Methods

Did the authors include a time limit on the duration of BUD cases that were included in the study?

Was the map an original source document created by the authors? There is no mention of this or the copyright material under Figure 2. The authors need to indicate the source of the map, if generated by them, the software used and its copyright need to be included in the manuscript.

On lines 156 -159, some percentages have been indicated and others in brackets. It is not clear what the authors intend to communicate by this, for instance “…main sources of water for drinking 20.9% (16.2% - 30.2%) and domestic activities 29.9% (26.5% - 37.8%) respectively”

It is interesting to note the numbers of BUD cases that were recruited in this study considering the study sites are not highly endemic districts and recruited cases were between July 2021 and March 2022.

Authors should describe how samples were collected and transported for cases that were laboratory confirmed for BUD. How was IS2404 PCR performed for these patients? At least a reference should be provided for the diagnostic PCR method used

Ethics

Authors need to indicate the age range they referred to as children in this study

On line 197, authors indicated the “data was stored securely” but it is not mentioned where exactly this information was stored; device, cabinet etc.

Data analysis

What is the origin of the STATA software on line 223?

Results

Titles of the Tables under the results section are long. Authors should revise the titles accordingly and exclude the addition of extraneous information such as “in the Akwapim South, Akwapim North-Okere, Jasikan and Biakoye Districts of Ghana; Community matched case control study, July 2021 to March 2022”.

The median age reported in this study is far higher than what is known for cases from West Africa. Added to this is the fact that most lesions were located on the lower limbs (and were not PCR confirmed) which in this age group is more probable to be from other aetiologies such as venous ulcers. This should be acknowledged as a major limitation.

It is unclear how the household economic status was assessed and what the categorizations (low/average/ high) mean. This needs clarification in the methods section

Discussion

It is important for the authors to reorganize the discussion section to appropriately convey the findings of their study. Beyond a few sentences in the opening paragraph, there is a restatement of some results in paragraph 1. These should be discussed by comparing with previous studies and indicating the significance/ implications of the findings for BU control in Ghana and elsewhere.

Line 339-341: Which model are the authors referring to here. No such data are presented in the results section of the manuscript. It is unclear what variables were included etc. They cannot discuss results which have not been presented as it makes it unclear for readers and should be addressed.

Line 346-369: These are limitations. It will be best to present these well after the study’s results and implications are discussed.

The authors acknowledge “recall bias “as a limitation of this study but failed to indicate the disease duration of participants that were recruited in this work. For those who had inactive BUD, what was the duration of illness before they were recruited in the study. This is a key confounder that can influence study results.

The authors indicated controls were selected from the homes or neighbouring communities of the cases. It seems rather interesting to read on line 367 the “geographic area for control selection was expanded”. Could the authors have introduced bias into the study by recruiting beyond the proposed sites initially indicated in the methods section? Kindly explain the nuances here.

The authors mentioned the use of alcohol as a cleansing agent for the wounds. I am curious to know what type of alcohol they are referring to here

PLOS authors have the option to publish the peer review history of their article (what does this mean? ). If published, this will include your full peer review and any attached files.

**Do you want your identity to be public for this peer review?** For information about this choice, including consent withdrawal, please see our Privacy Policy .

Reviewer #1: **Yes: ** MAMAN Issaka

Reviewer #2: No

**Figure resubmission:**

**Reproducibility:**



---

## [Decision Letter · Decision Letter 1]

22 Oct 2025

Response to Reviewers
Revised Manuscript with Track Changes
Manuscript

Shaden Kamhawi

co-Editor-in-Chief

Paul Brindley

co-Editor-in-Chief

**Reviewers' comments:**

**Key Review Criteria Required for Acceptance?**

**Methods:**

-Are the objectives of the study clearly articulated with a clear testable hypothesis stated?

-Is the study design appropriate to address the stated objectives?

-Is the population clearly described and appropriate for the hypothesis being tested?

-Is the sample size sufficient to ensure adequate power to address the hypothesis being tested?

-Were correct statistical analysis used to support conclusions?

-Are there concerns about ethical or regulatory requirements being met?

Reviewer #1: The author has implemented all recommendations raided during the reviewing.

Reviewer #2: (No Response)

**Results:**

-Does the analysis presented match the analysis plan?

-Are the results clearly and completely presented?

-Are the figures (Tables, Images) of sufficient quality for clarity?

Reviewer #1: Results are more accurate and clearly presented. The percentage of BU positive cases through the PCR is consequent and could help statistical analysis.

Reviewer #2: (No Response)

**Conclusions:**

-Are the conclusions supported by the data presented?

-Are the limitations of analysis clearly described?

-Do the authors discuss how these data can be helpful to advance our understanding of the topic under study?

-Is public health relevance addressed?

Reviewer #1: The conclusion is based on the finding of the study

The limitations are clearly described and do not impact of the resultats as well the conclusions stated in tthis study.

The discussion is well done.

Reviewer #2: (No Response)

**Editorial and Data Presentation Modifications?**

Reviewer #1: No revision is recommended

The paper could be accepted

Reviewer #2: (No Response)

**Summary and General Comments:**

Reviewer #1: The study is addressing the reducing of BU cases in four district in Ghana which is relatively similar in most of African countries. To investigate thr risk factor of contrating the BU is important in this decline of BU cases and could help to review or maintain the control of the disease.

Reviewer #2: The authors have done nearly all the required revisions suggested in the first review round.

There are a few corrections that require attention as detailed below:

In Table 3, What is meant by ‘farm proximal to water bodies’ and ‘farm distal to water bodies’. What do ‘proximal’ and ‘distal’ imply in this situation?

Line 479-480: Conclusion: As stated in the results section, farming close/ near water bodies was the identified risk factor (not living near water bodies). This should be corrected appropriately.

PLOS authors have the option to publish the peer review history of their article (what does this mean? ). If published, this will include your full peer review and any attached files.

**Do you want your identity to be public for this peer review?** For information about this choice, including consent withdrawal, please see our Privacy Policy .

Reviewer #1: **Yes: ** Issaka Maman

Reviewer #2: No

**Figure resubmission:**

**Reproducibility:**To enhance the reproducibility of your results, we recommend that authors of applicable studies deposit laboratory protocols in protocols.io, where a protocol can be assigned its own identifier (DOI) such that it can be cited independently in the future. Additionally, PLOS ONE offers an option to publish peer-reviewed clinical study protocols. Read more information on sharing protocols at https://plos.org/protocols?utm_medium=editorial-email&utm_source=authorletters&utm_campaign=protocols

---

## [Editor Report · Decision Letter 2]

24 Oct 2025

Dear Mr. Gohoho,

We are pleased to inform you that your manuscript 'Risk Factors for Buruli Ulcer Disease in Ghana: A Matched Case-Control Study in Four Selected Endemic Districts of Eastern and Oti Regions' has been provisionally accepted for publication in PLOS Neglected Tropical Diseases.

Best regards,

Michael Marks

Academic Editor

Mathieu Picardeau

Section Editor

Shaden Kamhawi

co-Editor-in-Chief

Paul Brindley

co-Editor-in-Chief

---

## [Editor Report · Acceptance letter]

Dear Mr. Gohoho,

We are delighted to inform you that your manuscript, "Risk Factors for Buruli Ulcer Disease in Ghana: A Matched Case-Control Study in Four Selected Endemic Districts of Eastern and Oti Regions," has been formally accepted for publication in PLOS Neglected Tropical Diseases.

Best regards,

Shaden Kamhawi

co-Editor-in-Chief

Paul Brindley

co-Editor-in-Chief
